# Extracellular Alterations in pH and K+ Modify the Murine Brain Endothelial Cell Total and Phospho-Proteome

**DOI:** 10.3390/pharmaceutics14071469

**Published:** 2022-07-15

**Authors:** Jared R. Wahl, Anjali Vivek, Seph M. Palomino, Moyad Almuslim, Karissa E. Cottier, Paul R. Langlais, John M. Streicher, Todd W. Vanderah, Erika Liktor-Busa, Tally M. Largent-Milnes

**Affiliations:** 1Department of Pharmacology, University of Arizona, Tucson, AZ 85721, USA; wahljare@email.arizona.edu (J.R.W.); anjali.vivek@creighton.edu (A.V.); sephmp2@email.arizona.edu (S.M.P.); malmuslim@email.arizona.edu (M.A.); jstreicher@email.arizona.edu (J.M.S.); vanderah@email.arizona.edu (T.W.V.); erikal@email.arizon.edu (E.L.-B.); 2BioIVT, ADME Product and Application Development, Old Westbury, NY 11568, USA; karissa.cottier@gmail.com; 3Department of Medicine, Division of Endocrinology, University of Arizona, Tucson, AZ 85721, USA; langlais@arizona.edu

**Keywords:** blood–brain barrier, pH, proteome, phospho-proteome, endothelial

## Abstract

Pathologies of the blood–brain barrier (BBB) have been linked to a multitude of central nervous system (CNS) disorders whose pathology is poorly understood. Cortical spreading depression (CSD) has long been postulated to be involved in the underlying mechanisms of these disease states, yet a complete understanding remains elusive. This study seeks to utilize an in vitro model of the blood–brain barrier (BBB) with brain endothelial cell (b.End3) murine endothelioma cells to investigate the role of CSD in BBB pathology by characterizing effects of the release of major pronociceptive substances into the extracellular space of the CNS. The application of trans-endothelial electrical resistance (TEER) screening, transcellular uptake, and immunoreactive methods were used in concert with global proteome and phospho-proteomic approaches to assess the effect of modeled CSD events on the modeled BBB in vitro. The findings demonstrate relocalization and functional alteration to proteins associated with the actin cytoskeleton and endothelial tight junctions. Additionally, unique pathologic mechanisms induced by individual substances released during CSD were found to have unique phosphorylation signatures in phospho-proteome analysis, identifying Zona Occludins 1 (ZO-1) as a possible pathologic “checkpoint” of the BBB. By utilizing these phosphorylation signatures, possible novel diagnostic methods may be developed for CSD and warrants further investigation.

## 1. Introduction

Changes in the extracellular milieu of the brain are reported during neurological disorders, including low pH and high extracellular potassium [1,2]. For example, cortical spreading depression (CSD) is a self-propagating wave of depolarization that spreads across the cerebral cortex, followed by hyperpolarization [3,4,5,6]. CSD events are associated with a number of neurological disorders, including migraine, traumatic brain injury (TBI), stroke, epilepsy, and multiple sclerosis (MS), making it a useful model with which to study the broad-spectrum changes at the blood endothelial barrier (BEB) associated with neurological disease states [7,8,9]. The contents of interstitial fluid during CSD are estimated to be ~68 μM glutamate, 60 mM K+, with pH fluctuating between 0.05 and 1.5 pH units away from physiologic (7.4) in both acidic and basic directions at the CSD wave front [10,11]. These elevations in glutamate and K+ are thought to propagate the spread of the CSD event by depolarizing nearby neurons; acidic spreading depression has been observed to occur in tandem with CSD [12,13,14,15]. In addition to neuronal and astrocytic activation [16,17], changes in the parenchymal extracellular composition are reported to influence the blood–brain barrier (BBB) and neurovascular unit (NVU) [12,18,19], though controversy remains [5,6,19,20,21,22,23]. 

The BBB is comprised of cerebral capillary endothelial cells, locked together by tight junction (TJ, apical side) and adherens junction (AJ, basolateral side) proteins to create a dynamic, highly selective permeable barrier between the blood and CNS [24,25,26]. Dynamic interactions of these junctional proteins allow for discrete regulation of substance influx and efflux to the brain [6,24,26,27,28]. These dynamic interactions can be deduced by examining interactions between functional residues within these proteins. Claudin-5 (CL-5), a critical mediator of BBB integrity, is a 23 kilodalton (kDa) protein of the Claudin multigene family, which interacts with neighboring extracellular CL-5 loops in the apical space of tight junctions. A conserved set of cysteine residues modulate contact adhesion allowing for a dynamic seal to form between endothelial cells [29]. In association with other proteins, including Occludin (OCC) and Zona Occludins-1 (ZO-1), the actin cytoskeleton is structurally linked to neighboring endothelial cells, allowing the formation of a dynamic, highly selective barrier to the CNS [29]. Using an in vitro model of the murine BEB with bEnd.3 cells, we investigate the effect of high parenchymal K+ and acidic pH (6.8) on the paracellular integrity, expression, and localization of tight junction proteins, and determine the global, as well as phospho-, and total proteome following these manipulations. Our findings suggest that high concentrations of both K+ and H+ ions within the abluminal extracellular environments of the CNS significantly change the function of the BEB by altering the dynamic regulation of the global and phosho-proteome of b.End3 cells. 

## 2. Materials and Methods

### 2.1. Cell Culture

b.End3 murine immortalized endothelial cells (ATCC, CRL-2299) were cultured under sterile conditions in 75 mm^2^ standard flasks (VWR, 10062-860) with Dulbecco’s modified eagle media (DMEM) (Gibco, 11995-065). DMEM was supplemented with 2 μM L-glutamine (Thermo Scientific, Waltham MA, USA; 25030081), 10% fetal bovine serum (FBS) (Gibco, 10082139), and 1% penicillin-streptomycin (Invitrogen, Waltham MA, USA; 15140122: (10,000 U penicillin)). C8-D1A murine astrocytes (ATCC, Manassas VA, USA; CRL-2541) were cultured in 75 mm^2^ standard flasks with DMEM supplemented with 10% (FBS) (Gibco, 10082139) and 1% penicillin-streptomycin (10,000 U penicillin). Both cell lines were split upon reaching 80% confluence to prevent overgrowth. Cell culture flasks were then incubated in a 37 °C humidified incubator with 5% CO_2_:95% air atmospheric conditions. 

### 2.2. Cell Treatments

b.End3 endothelial cells utilized in these studies were cultured in astrocyte conditioned media (ACM) for at least 24 h prior to any type of treatment, collection, or fixing. The usage of ACM for endothelial co-culture was integral to the functional culture of an endothelial barrier by supplying the critical growth factors and modeling the in vivo critical role of astrocytes in the maintenance of proper endothelial barrier function. ACM was produced in house by culturing fresh DMEM (Gibco 11995-065) cell culture medium supplemented with FBS (Gibco, 10082139) and penicillin 100 UI/mL-streptomycin 100 μg/mL (Invitrogen, 15140122) for 24 h with a confluent growth of C8-D1A mouse astrocytes. This media was aliquoted and frozen at −20 °C for use when needed. Culturing endothelial cells in ACM allows for the formation of a functional cell monolayer and tight junctions in vitro. To model a CSD event, cells were treated with a 5 min pulse of one of the following: (1) artificial cerebrospinal fluid (aCSF) (H_2_O, 148.19 mM NaCl, 3 mM KCl, 1.85 mM CaCl_2_, 1.71 mM MgCl_2_ 1.80 mM NaHPO_4_, 229.20 µM NaH_2_PO_4_) in ACM at equivolume to 60 mM KCl was used as a control (vehicle); (2) ACM buffered to a pH of 6.8 to model release of H+ ions into the extracellular space at the CSD wave front, prepared by titrating ACM down to a pH of 6.8 ± 0.05 with 12 M HCl (EM Sciences, Hatfield, PA, USA; HX0603-4); (3) glutamate (Thermo Scientific, A15031.36) dissolved in ACM at concentrations of 10 μM, 30 μM, and 100 uM; and (4) 60 mM KCl (Sigma Aldrich, P9541) dissolved in ACM, serving as a positive control. A total of 60 mM KCl treatments at relevant physiological levels were utilized as it is a typical condition used to evoke K+-ion-triggered spreading depolarization in live brain slices [30]. 

### 2.3. Transwell Cell Co-Cultures

In vitro modeling of the BBB was performed on a Transwell monoculture system, utilized for TEER, ^14^C-sucrose, and fluorescein isothiocyanate (FITC) uptake assays [31]. b.End3 endothelial cells were seeded on the luminal side of either 12- or 24-well Transwell inserts (Corning, 3460/3470) pretreated with 20% calf collagen. b.End3 cells were seeded on inserts and incubated for 48 h, following which abluminal wells were replaced with ACM and incubated for 24 h, all incubation being performed in a 37 °C humidified incubator with 5% CO_2_:95% air atmospheric conditions. Upon the formation of a luminal monolayer on the Transwell insert, co-cultures were then used for downstream analyses.

### 2.4. Trans-Endothelial Electrical Resistance (TEER)

The TEER technique utilizes measured changes in electrical resistance between two chambers filled with an aqueous solution and separated by a cultured cell barrier, with increased electrical resistance indicative of increased barrier integrity, as the free flow of ions between the chambers is prevented by the cellular barrier, manifesting as an increased electrical resistivity due to loss of electrical conductance between the chambers. (Figure 1a). b.End3 endothelial cells were seeded on Transwell inserts pretreated with 20% calf collagen and grown until formation of a monolayer, after which abluminal media was replaced with ACM for 24 h. A baseline TEER measurement was obtained via the chopstick method with an EVOM2 TEER meter (WPI, 91799), after which abluminal wells were treated by removing ACM and replaced with one of the following 5 min treatment preparations: (1) vehicle (aCSF in ACM), (2) ACM buffered to pH = 6.8, (3) 60 mM KCl (Sigma, P9541-5006) in ACM, and (4) 100 μM glutamate in ACM. After 5 min pulse, treatments were removed and replaced with fresh ACM and a 360 min time course was initiated. TEER measurements were obtained at the following time points: baseline (pre-treatment), 0 min (right after termination of 5 min pulse), 10, 20, 30, 60, 120, 180, and 360 min. Each experiment was repeated in triplicate for *n* = 3.

### 2.5. Immunocytochemistry

We immersed 12 mm round glass coverslips (Fisher Scientific, 12-545-80P) in 70% ethanol for one hour and air-dried them for 30 min under a UV lamp in a fume hood. Dried coverslips were placed in a 12-well plate and treated with a 20% collagen solution for two hours. Collagen solution was then removed by vacuum suction, and b.End3 cells were aliquoted in 80 μL volumes of DMEM media and incubated at 37 °C until the formation of a monolayer, after which a 24 h ACM incubation was initiated. Cells were then washed in 1× PBS and treated with one of the following preparations: (1) vehicle (aCSF in ACM), (2) ACM buffered to pH = 6.8, or (3) 60 mM KCl in ACM. After treatment cells were washed with 1× phosphatate buffered saline (PBS, prepped in house), then fixed with a 1% paraformaldehyde solution, permeabilized with 0.2% Triton X-100 (Sigma-Aldrich, St Louis MO, USA; T8787) in 1× PBS for 10 min at room temperature, and blocked in a 10% bovine serum albumin (BSA) solution (Gold Bio, St Louis MO, USA; A420-520) with 0.1% Triton X100 for 1 h. Primary antibodies (Table 1) prepared in 10% BSA with 0.1% Triton X-100 were added to cells and incubated overnight at 4 °C. The following day coverslips were washed with 1× PBS and treated with Alexa Fluor^TM^ fluorescent secondary antibodies (Invitrogen, A-21206/A-10037; Table 1) for 1 h at room temperature. Phalloidin staining (Table 1) was performed by treating fixed and permeabilized cells with Alexa Fluor^TM^ 488 conjugated Phalloidin (Thermo Scientific, A12379; Table 1) suspended at 1:40 in 1× PBS for 20 min at room temperature. Coverslips with fixed cells were then mounted on microscope slides by inverting and placing on a Pro-Long Gold Antifade Mountant with DAPI (4′,6-diamidino-2-phenylindole) (Invitrogen, P36931; Table 1) on cleaned glass microscope slides and imaged on an ECHO fluorescent microscope followed by confocal microscopy. Experiments were replicated in triplicate for *n* = 4.

### 2.6. ^14^C-Sucrose Transport Assays

b.End3 cells were seeded on the luminal side of Transwell inserts pretreated with 20% calf collagen and incubated at 37 °C with 5% CO_2_:95% air to allow the formation of a cell monolayer, upon which abluminal media was replaced with ACM and another 24 h incubation initiated. After incubation treatment, the pulses consisted of the following: (1) vehicle (aCSF in ACM), (2) ACM buffered to a pH = 6.8, and (3) 60 mM KCl in ACM. Pulse treatments were aliquoted in triplicate into a new 24-well plate, with inserts transferred from the original culture plate to the treatment plate for 5 min of treatment. A 30 min time course was initiated, and abluminal media was collected at 5 and 30 min timepoints in new collection plates. Scintillation vials for radiolabel quantification were prepared with Optiphase Supermax cocktail (PerkinElmer, 6013119) to act as a suspension agent. Radiolabeled ^14^C-sucrose (PerkinElmer, NEC100XOO1MC) was prepared by suspending 100 µL stock ^14^C-sucrose in 10 mL of DMEM, and 50 µL of this preparation was assayed for the working range of a radioactive emission of 50,000 counts per minute (CPM) ± 15,000. Once prepared, ^14^C-sucrose suspension was added to the luminal side of each insert, which was then immediately subjected to an abluminal treatment pulse, then transferred to a collection plate containing abluminal ACM. Abluminal media from the 5- and 30 min collection plates was then collected and aliquoted into a 5 mL scintillation vial (RPI, 905-5051), placed on a scintillation counter, and allowed to run overnight to capture CPM values. Samples were run in triplicate for each condition for *n* = 4. All radioactive material was disposed of according to University of Arizona regulations (RAM Protocol #698).

### 2.7. Fluorescein Isothiocyanate-Dextran Transport Assays

b.End3 cells were seeded onto the luminal side of Transwell inserts pretreated with 20% calf collagen. Inserts were cocultured with abluminal ACM and incubated at 37 °C in 5%:95% air CO_2_ for 72 h until the formation of an endothelial cell monolayer on the insert. Upon the formation of the monolayer, 1000 µg/mL of 4 or 70 kDa fluorescein isothiocyanate-dextran^1^ (Sigma-Aldrich, 46944/46945, 4/70 kDa FITC hereafter) solution was prepared in DMEM. Once prepared, the FITC preparation was added to the luminal side of Transwell inserts and pulsed abluminally for 5 min with the following treatments: (1) vehicle (aCSF in ACM), (2) ACM buffered to pH = 6.8, and (3) 60 mM KCl in ACM. All abluminal treatment media was then removed and replaced with fresh ACM, and a 180 min time course for the experiment was initiated. Aliquots of 10 µL were then removed from abluminal wells of each insert at timepoints of 10, 20, 30, 60, 120, and 180 min, and diluted into 90 μL DMEM to allow for a working volume and placed into a black clear-bottom microplate for fluorescence reading. Fluorescence readings were obtained with a ClarioStar plate reader (BMG Labtech) at an excitation wavelength of λ = 483 nm and emission wavelength of λ = 530 nm. The experiment was repeated 4 times in triplicate for an overall *n* = 4.

### 2.8. Western Immunoblotting

b.End3 cells were seeded on 6-well culture plates and incubated at 37 °C with 5% CO_2_:95% air and grown to confluence. Growth media was then removed and replaced with ACM, and cells were incubated for an additional 24 h. Cells were removed, washed with 1× PBS, and treated with a 5 min pulse of either (1) vehicle (aCSF in ACM), (2) ACM buffered to a pH = 6.8, and (3) 60 mM KCl in ACM. Treatments were removed and cells were then washed in 1× PBS, and 200 μL of cell lysis buffer (20 mM tris-HCl pH = 7.4, 50 mM NaCl, 2 mM MgCl_2_ hexahydrate, 1% NP40, 0.5% sodium deoxycholate, 0.1% sodium dodecyl sulfate in H_2_O) containing 1% by volume of protease and phosphatase inhibitor cocktail (Bimake, B14002/B15002) was added to each well and harvested via cell scraping. Lysed cell material was then added to a new 1.7 mL microfuge tube and centrifuged for 10 min at 13,000× *g* at 4 °C. Supernatant was then transferred to a new 1.7 mL microfuge tube for protein quantification.

Protein quantification was performed using a Pierce BCA protein assay kit (Thermo Scientific, 23223). Quantification standards were prepared from a 2 mg/mL Pierce BSA Standard (Thermo Scientific, 23210) serially diluted for the generation of quantification curve. Albumin standards and cell lysate samples were pipetted into a microplate in duplicate, and working reagent was added to each sample, covered, and incubated at 37 °C for 30 min. Subsequently, an incubation plate was assayed for optical density (OD) on a BMG ClarioStar plate reader at λ = 562 nm. OD values for BSA standards were used to generate a quantification curve, which was used to calculate the total protein concentration in each sample. Samples were then diluted to a concentration of 1 μg/μL in lysis buffer with 5× Lane Marker Reducing Sample Buffer (Thermo Scientific, 39000) and Dithiothreitol (DTT) (Thermo Scientific, R0862). Samples were then frozen at −80 °C.

Subsequently, 12-well precast 10% polyacrylamide gels (BioRad, 5671033) were loaded into a criterion gel electrophoresis cell, then filled with sodium dodecyl-sulfate polyacrylamide gel electrophoresis (SDS-Page) running buffer (BioRad, 1610722). Wells were then loaded with 30 μg of previously prepared cell lysate samples in triplicate by treatment group. An electrophoresis cell was run at 150 V for 10 min, then 190 V for 40 min. Following electrophoresis, gels were removed from their plastic cassettes and placed into a membrane transfer apparatus with a nitrocellulose membrane (Amersham, 10600001) and filled with SDS transfer buffer (BioRad, 1610771). The voltage was set at 20 V and transfer was run for 120 min. Following the transfer, apparatus was disassembled and the nitrocellulose membrane was removed and washed 3 times in 1× tris-buffered saline with 0.1% Tween^®^ 20 Detergent (TBST, prepared in house) for 5 min, then blocked in 5% milk in 1× tris-buffered saline (TBS, prepared in house) for 30 min. Primary antibodies were then added to the membrane (prepared in 5% BSA in 1× TBS), and the membrane was incubated at 4 °C for 48 h on a rocker. Following incubation, primary antibodies (Table 1) were removed, and membranes were incubated for 60 min with fluorescent secondary antibodies (LiCor, 926-68020/926-32211; Table 1) at room temperature. Secondary antibodies were removed and membranes were imaged on an Azure Sapphire Imaging machine. Blots were analyzed and quantified using UnScan It software (Silk Scientific).

### 2.9. Biotynylation

b.End3 cells were grown until confluence, after which they were incubated with ACM for 24 h. ACM was removed following incubation and cells were treated for 5 min with the following: (1) vehicle (aCSF in ACM), (2) ACM buffered to pH = 6.8, and (3) 60 mM KCl in ACM. Treatments were removed, and cells were washed in 1× PBS at pH 8.0 and chilled to 4 °C. NHS-SS (Succinimidyl-2-(biotinamido)-ethyl-1,3-dithiopropionate) biotin linking reagent buffer from a Thermo Cell Surface Isolation Kit (Thermo Scientific, 89881) was then added to cells and incubated at 4 °C for 25 min. A fresh aliquot of biotin buffer was added, and cells were incubated at 4 °C for another 25 min. Following incubation, the biotin buffer was removed, and cells were washed with pH = 8.0 buffered 1× PBS. Cell lysis buffer was prepared in house with protease and phosphatase inhibitors added at a ratio 1:100 and added to cells. Cell lysate was then scraped off the culture plates and transferred to a 1.7 mL microfuge tube and incubated on ice for 1 h. Subsequently, incubation tubes were centrifuged at 14,000 RPM for 10 min at 4 °C and supernatant was pooled by treatment group into a new 1.7 mL centrifuge tube. Neutravidin beads from the Cell Surface Isolation Kit were then equilibrated in lysis buffer and added to the supernatant previously collected. Tubes containing supernatant and beads were then sealed with parafilm and incubated overnight at 4 °C on a rocker to allow for mixing. The following day, each tube with was re-aliquoted into a new 1.7 mL microfuge tube and centrifuged at 14,000 RPM for 5 s to pellet Neutravidin beads. After removing and freezing supernatant, Neutravidin beads were washed following the following protocol at 4 °C: (1) three 3 min washes with 500 μL of lysis buffer, (2) two 3 min washes with a high-salt buffer (500 mM NaCl, 5 mM EDTA, 50 mM tris-HCl pH = 7.5 in H_2_O), and (3) one 3 min wash with 500 µL no-salt buffer (10 mM tris-HCl pH = 7.5 in H_2_O). Pellets were then eluted into 60 µL 2× Laemmli dye (BioRad, 1610737) with 10% DTT, and heated to 95 °C for 10 min, cooled, and loaded onto a 10% SDS-PAGE gel for electrophoresis, then placed into a transfer apparatus with a nitrocellulose membrane for the transfer of proteins to the membrane. Following the transfer, the membrane was blocked in 5% milk, and incubated with anti CL-5 antibody at 1:500 (Invitrogen, 35-2500; Table 1), anti VE-Cadherin (VE-CAD) 1:500 (Invitrogen, 36-1900; Table 1), and anti α-Tubulin 1:10,000 (Cell Signaling, 3873S; Table 1) primary antibodies for 48 h at 4 °C on a rocker. Following primary antibody incubation, membrane was treated with secondary anti-rabbit and anti-mouse fluorescent secondary antibodies (Table 1) at a concentration of 1:10,000 for 1 h at room temperature. Membranes was then imaged on a Licor fluorescent imaging apparatus, and bands were analyzed as explained in the Western Immunoblotting procedure. Experiments were performed in triplicate per condition for *n* = 2.

### 2.10. Proteomics

b.End3 murine endothelial cells were treated with either a 24 h hormone treatment or pulsed with one of the CSD constituent substances, as previously described. b.End3 cells were then lysed post treatment and loaded into SDS-PAGE for separation by electrophoresis. A total of 200 μg of harvested cell lysate supernatant was separated on a 10% SDS-PAGE gel and stained for total protein presence with Bio-Safe Coomassie G-250 Stain. Lanes from the gel were separated and cut into six slices, which then underwent trypsin digestion, and resulting peptides were purified by C18 desalting performed as described by Kruse et al. [32]. High-performance liquid chromatography-electrospray ionization tandem mass spectrometry (HPLC-ESI-MS/MS) was performed in positive-ion mode on an Orbitrap Fusion Lumos tribrid mass spectrometer (Thermo Scientific, IQLAAEGAAPFADBMBHQ) fitted with an EASY-spray source (Thermo Scientific, ES081). NanoLC was performed according to the protocol published by Kruse et al. [32]. Tandem mass spectra were extracted from files in Xcalibur ‘RAW’, and ProteoWizard 3.0 msConvert script was used to assign charge states with default parameters. Mascot (Matrix Science, ver 2.6.0) software was used with default probability cut-off score settings to search fragment mass spectra against the *Mus musculus* database in SwissProt_2018_01 (16,965 entries). The search variables used were as follows: 10 ppm mass tolerance for precursor ion masses, and 0.5 Da for product ion masses, trypsin digestion, maxima of two missed tryptic cleavages, variable modifications of phosphorylation of threonine, tyrosine, and serine, and oxidation of methionine. Scaffold software (Proteome Software, version 4.8.7) was used to cross-correlate Mascot search results with X! Tandem software. Significance value was set at *p* ≥ 0.05. Ion intensity-based label-free quantification was performed using Progenesis QI for proteomics software (Nonlinear Dynamics, version 2.4). Raw files were imported and converted into two-dimensional maps with *y* axis defined as time and *x* axis defined as *m/z*, which was then followed by the selection of a reference run for alignment. The aligned runs were then used to create an aggregate data set containing all peak information from all samples, after which the data pool was narrowed down to only +2, +3, and +4 charged ions for further analyses. The top 8 most intense precursors of a given feature were grouped into a peak list of fragment ion spectra and exported in a Mascot generic file (.mgf) and searched against the *Mus musculus* SwissProt_2018_01 database utilizing Mascot software. The following search variables were used: 10 ppm mass tolerance for precursor ion masses and 0.5 Da for product ion masses, trypsin digestion, maxima of two missed tryptic cleavages, variable modifications of oxidation of methionine, and phosphorylation of serine, tyrosine, and threonine, ^13^C = 1. The data were collected into a Mascot .xml file and imported into Progenesis allowing for the assignment of peptides and proteins. Peptides with a Mascot ion score < 25 were not used for further analyses. Non-conflicting peptides and precursor ion abundance values were normalized using a reference run to perform protein quantification. A heat map of principal component analyses (PCAs) and unbiased hierarchal clustering analyses were performed in Perseus [33,34].

### 2.11. Phospho-Proteomics

b.End3 cells were cultured and treated as previously described. A total of 5 mg of protein lysate per sample (*n* = 4) underwent tryptic digestion and enrichment of phosphopeptides with sequential enrichment from metal oxide affinity chromatography, as per the manufacturer’s instruction (Thermo Scientific). A Thermo Orbitrap Fusion Lumos Tribrid Mass Spectrometer fitted with an EASY Spray source was used to perform HPLC-ESI-MS/MS in positive-ion mode, according to the manufacturer’s protocol. NanoLC was performed using a Thermo Scientific UltiMate 3000 RSLCnano System with an EASY Spray C18 liquid chromatography column (Thermo Scientific, 50 cm 3 75 mm inner diameter, packed with PepMap RSLC C18 material, 2 mm, cat. # ES803); loading phase for 15 min at 0.300 mL/min; mobile phase, linear gradient of 1% to 34% buffer B in 119 min at 0.220 mL/min, followed by a step to 95% buffer B over 4 min at 0.220 mL/min, hold 5 min at 0.250 mL/min, and then a step to 1% buffer B over 5 min at 0.250 mL/min and a final hold for 10 min (total run 159 min); buffer A 5 0.1% formic acid/H_2_O; and buffer B 5 0.1% formic acid in 80% acetonitrile. All solvents utilized were liquid chromatography mass spectrometry grade. Xcalibur software (Thermo Scientific, version 2.3) was used to acquire Spectra, and Progenesis QI (Nonlinear Dynamics, version 2.4) was used to perform ion-free intensity-based label-free quantification. .raw files were imported and converted into two-dimensional maps with *y* axis defined as time and *x* axis defined as *m/z*, which was then followed by a selection of a reference run for alignment. The aligned runs were then used to create an aggregate set containing all peak information from all samples, after which the data pool was narrowed down to only +2, +3, and +4 charged ions for further analyses, which were then grouped by treatment. A peak list of fragment ion spectra was generated and exported in Mascot generic file (.mgf) and searched against the *Mus musculus* SwissProt_2018_01 database, utilizing Mascot software with the following search variables: 10 ppm mass tolerance for precursor ion masses and 0.5 Da for product ion masses, trypsin digestion, maxima of two missed tryptic cleavages, variable modifications of oxidation of methionine, and phosphorylation of serine, tyrosine, and threonine, ^13^C = 1. Protein or peptide assignment was performed by importing the resultant Mascot .xml file into Progenesis, while peptides with a <25 Mascot ion score were not used further. Precursor ion abundance values for peptide ions were normalized to all proteins. Differences were assessed as significant if a difference between vehicle and treatment groups was *p* < 0.05 assessed with one-way analysis of variance (ANOVA). Consensus phosphorylation sequences were determined using iceLogo [35,36,37]. Heat maps and PCA analyses were performed in Progenesis.

### 2.12. Statistics

Numbers required to achieve statistical power for assays below were determined a priori in G.Power 3.1 in alignment with the National Institute of Health (NIH) policy (NOT-OD-15-102), so that differences of 20% were detected with 80% power at a significance level of 0.05. Post-experimental data analyses were performed in GraphPad Prizm 7.0 (GraphPad Software). Unless otherwise noted, data were analyzed with either a two-way paired or unpaired *t*-test, or one-way ANOVA with either Bonferroni, Dunnett, or Tukey test administered *ad hoc.*

## 3. Results

### 3.1. In Vitro Modeling and Functional Significance of CSD Induced Paracellular Leak in BEB

Pronociceptive substances released at the wavefront of CSD events were screened to qualitatively assess the significant induction of a paracellular leak in the BEB. Substances found significant were then assayed for their functional impact on BEB with quantitative transport assays to assess the magnitude of the BEB breach and functional outcomes on BEB.

#### 3.1.1. TEER Screening of BEB Demonstrates Rapid Induction of Paracellular Leak by Abluminal K+ and H+ Ion Treatment in a Transwell In Vitro Model of the BEB

The integrity of the BEB was assayed via TEER (Figure 1a,b) to ascertain the individual ability of each substance to induce a breach in the BEB. Following a 120 min time-course TEER screening demonstrated a rapid, significant drop in electrical resistance immediately following 5-min basolateral treatment with 60 mM KCl (** *p* = 0.0010) and acidified media (** *p* = 0.0011), (Figure 1b), maintaining low TEER values throughout the 120 min time course. Treatment with 100 μM glutamate (*p* = 0.2941) and 100 μM ATP (** *p* = 0.0058) demonstrated no significance, and a gradual drop in TEER became significant 20 min following treatment, respectively. The rapid reduction in TEER values following KCl and acidified pH delayed the effect of ATP, and no effect from glutamate suggests a fast-acting mechanism regulating BEB functionality.

#### 3.1.2. Functional BEB Integrity Is Disrupted by KCl, Acidified pH, and ATP, but Not High Concentrations of Abluminal Glutamate

The rapid drop in TEER following KCl and acidified pH treatment observed in the TEER screenings warranted further investigation into the functional consequences on the BEB integrity following these insults. Utilizing the same Transwell culture model, ^14^C-sucrose, 4 and 70 kDa FITC uptake assays were performed over a 30 and 180 min time course, respectively, to assess the functional alterations to the BEB as well as quantifying the magnitude of the BEB breach. Both 4 and 70 kDa FITC uptakes demonstrated no significant differences between treatments (Figure 1e,f), and this was most likely due to the higher molecular weights of the fluorescent markers. Interestingly, ^14^C-sucrose at a molecular weight of 300 Da showed highly significant abluminal uptake immediately following KCl (** *p* = 0.009) and acidified pH treatment (**** *p* < 0.0001) (Figure 1c), which was then abolished back to baseline after 30 min (Figure 1d). This observation would suggest that KCl and elevated H+ treatment induce a breach permissive of no greater than 300 Da, and a rapid, dynamic reannealing of the BEB. These data also indicate no loss of functionality to the proteins comprising the tight junctions of the BEB following KCl and H+ insult, suggesting a reversible underlying mechanism. 

### 3.2. Changes in TJ Localization, but Not Total Expression, Are Induced by Mediators Released during CSD

Our previous in vivo studies indicated that the total detection of expressed TJ proteins CL-5 and OCC was unchanged in isolated microvessels following CSD induction [6]. To determine if the relocalization of the TJ and associated proteins underscored the rapid increase in and dynamic ablation of the paracellular leak in the BEB following CSD insult, confocal microscopy and immunofluorescent techniques were utilized to observe and quantify the changes to TJ protein localization following KCl and acidified pH treatment, as well as the induction of f-actin stress fiber formation. Alterations to the global b.End3 proteome following KCl or acidic pH insult were then assessed with quantitative proteomics.

#### 3.2.1. Confocal Immunofluorescence and Quantification of CL5, but Not ZO-1 and VE-CAD, Demonstrated a Significant Alteration to Localization Following KCl and Acidified pH Treatment

Alterations to the localization of the proteins CL-5, ZO-1, and VE-CAD were assayed to observe changes at the TJ (CL-5), AJ (VE-CAD), and intracellular compartment (ZO-1) of the endothelial cells following KCl and acidified pH insult (Figure 2a). Confocal images taken at 40× magnification for each treatment condition were quantified for corrected total cell fluorescence (CTCF) to assess alterations in the fluorescent signal of each protein post treatment (Figure 2b). When compared to media, both KCl and acidified pH induced significant alterations to CL-5 fluorescence, with KCl (* *p* = 0.0351) reducing and acidic pH (* *p* = 0.521) increasing CL-5 CTCF. These treatments had no significant effect on both ZO-1 and VE-CAD fluorescence. These observations suggest the rapid onset of apical protein relocalization in vitro.

#### 3.2.2. Confocal Imaging and Quantification of f-Actin Filaments Demonstrate Increase in f-Actin Stress Fibers Following KCl Insult

The structural integrity of the endothelial cell TJ complex is dependent on intracellular actin cytoskeletal linkage to transmembrane TJ proteins, such as CL-5 and OCC via ZO-1, 2, and 3. Cells undergoing structural insult can be identified by the increased detection of filamentous actin (F-actin) vs. normal globular (G-actin) actin. FITC-conjugated phalloidin was used to stain f-actin filaments following KCl and acidified pH treatment (Figure 2a,b). Following KCl insult, a significant increase in F-actin stress fibers was observed when compared to vehicle (* *p* = 0.0415), indicative of potential homeostatic stress response following KCl insult.

### 3.3. Dynamics of CL-5 Trafficking and Localization within Cellular Components

CL-5 has recently been validated as a clinical biomarker for several CNS disorders; therefore, focus was placed on this protein to further characterize its utility as a tool for investigating alterations to TJ integrity under experimental conditions, and to further characterize the potential of clinical diagnostic use. Biotynylation cell surface protein isolation assays were utilized to quantify the changes in surface localization of CL-5 following KCl and acidified pH insult. 

#### Endothelial Cell Surface Localization of CL-5 Was Significantly Increased Following Acidified pH Treatment, While VE-CAD Surface Detection Was Unaffected

CL-5 and VE-CAD serve as major endothelial cell tethering proteins within the apical TJ and basolateral AJ, respectively. The localization of both proteins following KCl and acidified pH insult was assayed via biotynylation isolation to assess relocalization post exposure and to deduce the primary location of effect to the apical or basolateral side of the cell. Exposure to acidified pH (* *p* = 0.0231) was shown to increase the surface detection of CL-5 vs. vehicle (Figure 3b), while VE-CAD was not altered following exposure to acidified pH or KCl (Figure 3a). These observations suggest that the effect of KCl exposure on cell surface protein trafficking is limited to the locality of physical contact with the cell within the short time frame of CSD. 

### 3.4. Mechanisms of CL-5 Reorganization

To determine the mechanisms underlying the relocalization of the TJ proteins and associated increase in BEB permeability, we next employed unlabeled quantitative total and phospho-proteomic analyses of b.End3 cells post KCl and acidified pH treatment. Total proteome analysis allows for the assessment of changes to the homeostatic environment of the cell through quantification of total protein enrichment, while phospho-proteomic analysis allows for the analysis of post-translational modifications (PTMs) to the global proteome. 

#### 3.4.1. Global Quantitative Unlabeled Proteome Analysis Demonstrated Significant, Differential Changes to Total Enrichment of TJ and Cytoskeletal Associated Proteins Following KCl and Acidified pH Exposure

Global proteome analysis of b.End3 cells post treatment detected a total of 7113 proteins, 6279 of which were identified as being statistically significant (*p* < 0.0544). Acidified pH downregulated 68 proteins and upregulated 160 proteins (Figure 4A,C), while 49 proteins were downregulated following KCl exposure (Appendix A). Comparison of total protein enrichment by each treatment group revealed no significant overlap in effect between KCl and acidic pH (Figure 4B), implying unrelated mechanism of effect on protein enrichment by treatment. Divergent effects on CL-5 and ZO-1 enrichment were observed following KCl and acidified pH exposure, with CL-5 being downregulated by acidic pH (*p* = 0.2286) and ZO-1 upregulated following KCl exposure (* *p* = 0.0207) (Figure 4D). Degradation of cell structural integrity was further observed in changes to expression of three proteins associated with actin cytoskeletal processes. These were found to be significantly altered following KCl and acidified pH treatment (Figure 4D). Actin filament associated protein 1 (AFAP1), F-actin capping protein subunit alpha 2 (Caza2), and F-actin capping protein subunit beta (Cap2b) are all necessary for actin function, and changes to expression could potentially induce a loss of structural integrity of the cell. Acidified pH treatment had a differential effect on all three proteins, upregulating AFAP1 (*p* = 0.0034) while downregulating Caza2 (*p* = 0.0006) and Caz2b (*p* = 0.0508). KCl exposure significantly downregulated Caza2 (*p* < 0.0001). Taken together, these data indicate that the loss of BEB integrity is due to a synergistic convergence of unrelated individual deleterious processes induced by KCl and acidified pH exposure to TJ proteins CL-5, ZO-1, and actin maintenance proteins AFAP1, Caza2, and Cap2b. 

#### 3.4.2. Functional and Pathway Analyses of Global b.End3 Proteome with GO and KEGG Bioinformatic Databases

Proteins showing statistically significant changes in enrichment from the total proteome analysis were analyzed in the Gene Ontology (GO) and Kyoto Encyclopedia of Genes and Genomes (KEGG) bioinformatics databases to assess the functional impact of protein expression changes detected in our analysis following KCl and acidified pH exposure. Separated by treatment group, significant proteins (α < 0.05) were queried through the following three GO bioinformatic databases: biological processes (BPs), cell compartment (CC), and molecular function (MF). Additionally, the KEGG pathway database was screened, and positive hits were scored by fold enrichment (FE), with the top ten from each database listed, separated by treatment group, and graphed by FE score against log transformed -Log10 *p*-values to visualize the functional effect on pathways and processes related to TJ and cytoskeletal homeostasis (Figure 5a,d). Database queries allowed for the identification of upstream proteins involved in the structural maintenance of the cell whose global enrichment levels were altered following KCl and acidic pH exposure (Figure 4D and Figure 6D). These observations imply the synergistic effect of individual mechanisms of KCl and acidic pH summing to an increased effect.

#### 3.4.3. Global b.End3 Phospho-Proteomic and Bioinformatic Analyses Identified Significant and Highly Variable Enrichment of Phosphorylation Sites on TJ and Cytoskeletal Proteins Uniquely Associated with Treatment

Phosphorylation of CL-5 is associated with increased permeability of the BBB, but it is unknown if CSD-like conditions induce these post-translational modifications (PTMs). Following treatment with vehicle, KCl, and acidified pH, cells were quantified for total phosphorylation enrichment and amino acid residue enrichment of phosphorylation sites by individual protein. A total of 25,546 total phosphorylation adducts were detected, of which 11,371 were deemed significant by 3-way ANOVA (α = 0.05). When analyzed by treatment group, 237 proteins experienced phosphorylation following vehicle treatment vs. KCl and acidic pH (Figure 6A). A total of 1074 proteins underwent phosphorylation enrichment after acidified pH exposure (Figure 6C), and KCl upregulated phosphorylation in 302 proteins (Appendix A). Global comparisons of proteins experiencing significant phosphorylation enrichment were compared by treatment group, and the overlap of total increased phosphorylated proteins was observed between treatments (Figure 6B); however, these co-targeted proteins were not assessed for specific residue enrichment by treatment group, and when factoring in the total quantity of phosphorylation sites, the comparison suggests that a difference between treatments is due to independent mechanisms of phosphorylation enrichment. Total proteome enrichment was compared to global phosphorylation enrichment to further assess this observation. The enrichment of both total expression and phosphorylation of proteins involved in structural maintenance pathways co-occurred within treatment groups (Figure 6D). This suggests varied mechanisms of structural maintenance pathway manipulation unique to each treatment. The functional analysis of phosphorylation enrichment queried through GO and KEGG databases further validate this finding. Similar to the findings with total proteome enrichment, each treatment group was found to have a unique effect on functional pathways in each database (Figure 7a–d). Interestingly, ZO-1 was observed to have a high degree of PTMs across all treatments, and the residues targeted for phosphorylation were highly varied and unique to each treatment (Figure 8b,d,f). This would suggest that ZO-1 may be a primary target for PTM-regulated control of TJ protein trafficking and localization, as the differential effects observed from each treatment in the previous experiments was congruent with the unique peptide phosphorylation enrichment signatures induced by each treatment. The function of ZO-1 as a link between the actin cytoskeleton and the transmembrane TJ proteins lends further support to the potential of ZO-1 acting as a major regulator of TJ protein trafficking and localization.

#### 3.4.4. Analysis of Phospho-Peptide Enrichment Demonstrates Unique Enrichment of ZO-1 by Treatment Group

Proteins were assessed for both total and specific phosphorylation-enriched residues on the peptide chains comprising their primary structure. Highly enriched proteins associated with TJ or cytoskeletal function were separated by treatment group and visualized by scatterplot comparing Log2 transformed FE of phosphorylation against the −Log10 transformation of *p*-values (Figure 8a,c,d). Proteins phosphorylated at residue positions verifiable in UniProt were then plotted by comparing the −Log2 transformation of their fold enrichment score to each iteration of a unique phosphorylated residue. Proteins were assessed for fold enrichment of phosphorylation per treatment and plotted by highest overall level of phosphorylation enrichment by treatment. The locations of phosphorylated residues are also shown.

## 4. Discussion

Increases in local extracellular K+, disruption of ionic gradients for Na^+^, Cl^−^, and H^+^, and flux of ATP and glutamate are reported during multiple neurological diseases and are reported to induce neurovascular decoupling [38,39,40,41] to open the BBB [6,22,23,42,43]. This study identified that high extracellular concentrations of K+, H+ acidification below pH 7.4, and ATP, but not glutamate, at concentrations reported during CSD events reduce TEER and induce paracellular permeability by relocalization through functional alteration to actin cytoskeletal and tight junctional protein complexes in murine brain endothelial cells. The determination of the global and phospho-proteomes identified unique signatures associated with high abluminal K+ as compared to acidic pH, suggesting that the functional outcomes of BEB integrity during neurologic diseases reflect the dynamic influence of each chemical mediator acting through unrelated physiologic pathways, whose effect is synergized following the temporal convergence of each mediator resulting in the amplification of deleterious neurologic functional outcomes.

Consistent with prior reports, we found that high extracellular K+, acidified pH, and ATP reduced TEER and functionally allowed the paracellular uptake of small molecules [44,45,46,47]. In contrast, glutamate at sub-physiological levels and above reported levels for CSD did not change TEER values, suggesting it did not contribute to BEB permeability under these conditions. This observation is at odds with the prior reports showing that glutamate at supraphysiologic concentrations (>100 μM–10 mM) reduced TEER and increased BBB permeability [48,49]. Notably, ATP induced rapid changes in TEER, but a delayed effect on functional BEB perturbations; these observations were consistent with those of Maeda et al. [44]. They proposed that ATP acts to increase the secretion of matrix metalloproteases, which have been implicated in BBB permeability during headaches [50]. Thus, the transient permeability of the BBB during CSD as reported is likely attributable to microenvironment levels of K+, pH, and ATP. 

Dysfunction at the BBB is reported as being correlated with the loss of TJ complex integrity via protein downregulation, relocalization within the plasma membranes, and phosphorylation, to name a few [14]. Two main classifications of TJ proteins exist: transmembrane proteins, such as the Claudins, Occludin, Cadherin, and Junctional Adhesion Molecules (JAMs), and anchoring proteins, such as the Zona Occludins (ZOs) family [15]. Previous research shows that Claudin-5 and Occludin expression do not change after KCl induced CSD on isolated microvessels, suggestive of a different underlying mechanism than downregulation contributing to permeability [6]. Data from immunofluorescence and biotinylated cell surface isolation analyses indicated that CSD-like concentrations of acidic pH changes induce plasma membrane reorganization of CL-5, but not VE-CAD and ZO-1. This would indicate the functional role of protein localization on the cell in determining a response to acidic pH insult. No change in CL-5 immunoreactivity was detected qualitatively in extracellular compartments post insult, suggesting trafficking of CL-5 post KCl and acidic pH insult is restricted to membrane and cytosolic compartments, further indicating that an alteration to protein localization, not expression, may play a role in our functional observations. Global proteomic analysis showed that pH significantly reduced the detection of CL-5 and KCl increased ZO-1 enrichment; VE-CAD expression was unchanged. In summary, these data suggest independent functional outcomes on TJ proteins unique to K+ and acidified pH insult, implying separate unrelated mechanisms. 

These proteomic data seem to contrast with our previous report [6]: the preparation used (i.e., ex vivo vs. in vitro, immortalized cells) and the isolated effects of acidified pH or K+ versus a composite application of both may underscore the differences. Endothelial transcytosis via a caveolin-1 regulated mechanism was reported by Sadeghian et al. to contribute to BBB permeability following CSD in an in vivo murine model, as opposed to TJ protein relocalization [51]. While the time course of Sadeghian et al. mirrored the investigations of Cottier et al. [6], the in vitro investigations undertaken here examined a much shorter time course and utilized much smaller permeability markers (^14^C-sucrose, 4 kDa FIC) as opposed to Evans Blue and the exclusive use of 70 kDa FITC. These differences in methodology may have contributed to the opposing findings contained herein. The variable detection of F-actin stress fibers following KCl and acidified pH insult was confirmed in the global proteomic analysis. The expression of actin-associated proteins AFAP1 and Caza2 and Cap2b were significantly altered following acidic pH exposure, and KCl heavily downregulated Caza2. Mirroring the treatment-dependent functional changes observed at the TJ, differential molecular mechanisms may underlie these changes. The findings from total proteome analyses following modeled CSD insult indicate a potential major homeostatic disruption to several key structural proteins in endothelial cells, evidenced by a significant alteration to the enrichment of actin-associated proteins AFAP1, Caza2, and Capzb, in addition to CL-5 and ZO-1 changes. The discrete manner by which each specific CSD constituent plays a role in these observations warrants further investigation into a potential PTM mechanism.

Data from this study demonstrate that acidic pH dynamically regulates more proteins and phosphorylation PTMs than KCl in murine brain endothelial cells centered on the downregulation of actin reorganization and capping, while upregulating responses to exocyst organization and negative regulation of membrane tubulation. Phosphorylation of proteins related to adheren junction’s reorganization and localization was also downregulated by acidified pH. KCl globally facilitated microtubule formation by downregulating depolymerization at the signaling level by upregulating actin formation pathways. This treatment-dependent effect was further validated by the absence of overlap in total protein enrichment between treatments, suggesting changes in brain pH and K+ concentration increase BEB permeability via distinct mechanisms. 

Phosho-proteomic analysis indicated PTMs as a likely mechanism for divergent treatment response, as phosphorylation is a reversable and highly dynamic mechanism of protein regulation. ZO-1, a major linking protein between the cytoskeleton and TJ was significantly enriched in phosphorylated residues specific to each treatment resulting in similar functional outcomes. This functional convergence on ZO-1 from both treatments may also indicate a central regulatory role for ZO-1 in response to CSD-like insult; specifically when considering the critical function, it performs maintaining structural homeostasis. Therefore, similar functional outcomes due to divergent KCl and acidified pH phosphorylation sites on ZO-1 demonstrate a potential mechanism driving protein relocalization and the paracellular leak of the BEB.

## 5. Conclusions

Low brain pH and high extracellular K+ during neurological disorders promote BEB/BBB paracellular leak by inducing TJ and actin cytoskeletal reorganization and functional alteration through independent mechanisms. The elucidation of these molecular mechanisms may be utilized as a “diagnostic fingerprint” specific to a unique pathology potentially useful as a biomarker.

## Figures and Tables

**Figure 1 pharmaceutics-14-01469-f001:**
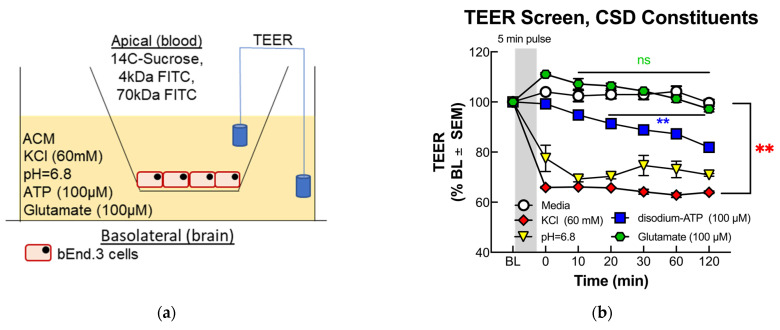
Screening and functional consequences on BEB following treatment with pronociceptive substances released during CSD. (**a**) Cross-sectional diagram of Transwell culture and TEER assay. (**b**) Both 60 mM KCl and acidified pH induced a rapid, significant drop in TEER following abluminal treatment, while 100 μM ATP demonstrated a gradual TEER drop, and 100 μM glutamate had no significant effect (60 mM KCl vs. Veh: mean of differences: −32.5 ± 5.463, ** *p* = 0.0010; *n* = 3/4: glutamate vs. Veh: mean of differences = 1.612 ± 1.403, *p* = 0.2941, *n* = 3: pH = 6.8 vs. Veh: mean of differences = −25.77 ± 4.392, ** *p* = 0.0011, *n* = 3/4: 100 μM ATP vs. Veh: mean of differences = −10.39 ± 2.483, ** *p* = 0.0058, *n* = 3: all data analyzed with two-tailed paired *t*-test ± SEM). Abluminal ^14^C-sucrose uptake by treatment group at (**c**) 5 and (**d**) 30 min post treatment. Both 60 mM KCl and acidified pH induced significant increase in ^14^C-sucrose uptake 5 min post-treatment, (Veh vs. pH = 6.8: mean difference = −52.29; 95% CI (−73.35, −31.22) **** *p* = < 0.0001, *n* = 8: Veh vs. 60 mM KCl: mean difference = −27.21; 95% CI (−48.27, −6.139) ** *p* = 0.009, *n* = 8; analyzed by one-way ANOVA with Dunnett’s multiple comparison test post hoc F(3,29) = 12.79), which ablated after 30 min (Veh vs. pH = 6.8: mean difference = −10.15; CI (−29.65, 9.356) *p* = 0.4440, *n* = 8: Veh vs. 60 mM KCl: mean difference = −7.825; 95% CI (−7.825, 30.02) *p* = 0.3499, *n* = 9, ns = no significance: analyzed by one-way ANOVA with Dunnett’s multiple comparison test post hoc. F(3,31) = 2.554). (**e**) Both 4 and (**f**) 70 kDa FITC uptake assays demonstrated no significant difference between treatments over a 180 min time course, suggesting magnitude of BEB breach is both <4 kDa (Veh vs. pH = 6.8: mean difference = −698,151; 95% CI (−14,413,073, 13,016,771) *p* = 0.9984, *n* = 4: Veh vs. 60 mM KCl: mean difference = −395,731 95% CI (−14,110,653, 13,319,191) *p* = 0.9997, *n* = 4: analyzed by one-way ANOVA with Dunnett’s multiple comparison test post hoc; F(3,20) = 0.2069, ns = no significance) and 70 kDa (Veh vs. pH = 6.8: mean difference = −524,392; 95% CI (−4,332,275, 3,283,491) *p* = 0.9713, *n* = 4: Veh vs. 60 mM KCl: mean difference = 38,393; 95% CI (−3,769,490, 3,846,276) *p* = 0.9999, *n* = 4: analyzed by one-way ANOVA with Dunnett’s multiple comparison test post hoc; F(3,20) = 0.06026, n.s. = no significance). The green line in (**e**) is an error bar for the pH data.

**Figure 2 pharmaceutics-14-01469-f002:**
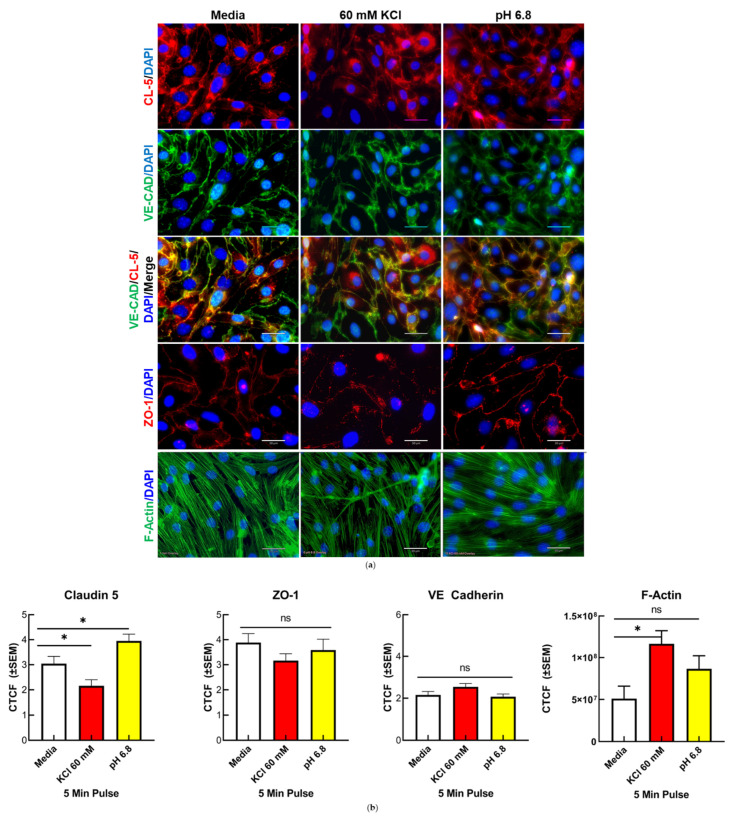
Localization of endothelial TJ associated proteins and visualization of F-actin fibers following KCl and acidified pH treatment in vitro. (**a**) Representative immunofluorescence images of b.End3 endothelial cells post 60 mM KCl and acidified pH treatment for CL-5, VE-CAD, ZO-1, and F-Actin, scale bar represents 30 μm (**b**) Quantification of CL-5, ZO-1, VE-CAD, and F-Actin by corrected total cell fluorescence (CTCF). CL-5, but not ZO-1 or VE-CAD, demonstrated significant changes to total signal following KCl and acidified pH treatment, while F-actin was increased following KCl insult. CL-5: media vs. 60 mM KCl: mean rank difference = 25.20, Z = 2.107, * *p* = 0.0351, n1 = 34, n2 = 43: media vs. pH = 6.8: mean rank difference = −21.48, Z = 1.942, * *p* = 0.0521, n1 = 34, n2 = 64; Kruskal–Wallis statistic: 36.25: ZO-1: media vs. 60 mM KCl: mean rank difference = 12.77, Z = 1.319, *p* = 0.1872, n1 = 25, n2 = 33; media vs. pH = 6.8: mean rank difference = 9.352, Z = 0.9384, *p* = 0.3481, n1 = 25, n2 = 29; 6.382; Kruskal–Wallis statistic: 6.382, *p* summary: 0.0945. VE-CAD: media vs. 60 mM KCl: mean rank difference = −16.30, Z = 1.312, *p* = 0.1894, n1 = 29, n2 = 46: media vs. pH 6.8: mean rank difference = 5.375, Z = 0.4411, *p* = 0.6592, n1 = 29, n2 = 51, ns = no significance, Kruskal–Wallis statistic: 10.12, * *p* summary: 0.0175. Analyzed by Kruskal–Wallis test with post hoc uncorrected Dunn’s test. F-actin: media vs. pH 6.8: mean difference = −35,648,452, 95% CI (−98,074,135, 26,777,231), n1 = 3, n2 = 3, *p* = 0.2510, n.s.= no significance; media vs. 60 mM KCl: mean difference = −65,606,966, 95% CI (−128,032,649, −3,181,283), n1 = 3, n2 = 3, * *p* = 0.0415. F(2,6) = 4.538; *p* = 0.630). All measurements were calculated as integrated fluorescence density—(individual cell area * mean background fluorescence) ± SEM.

**Figure 3 pharmaceutics-14-01469-f003:**
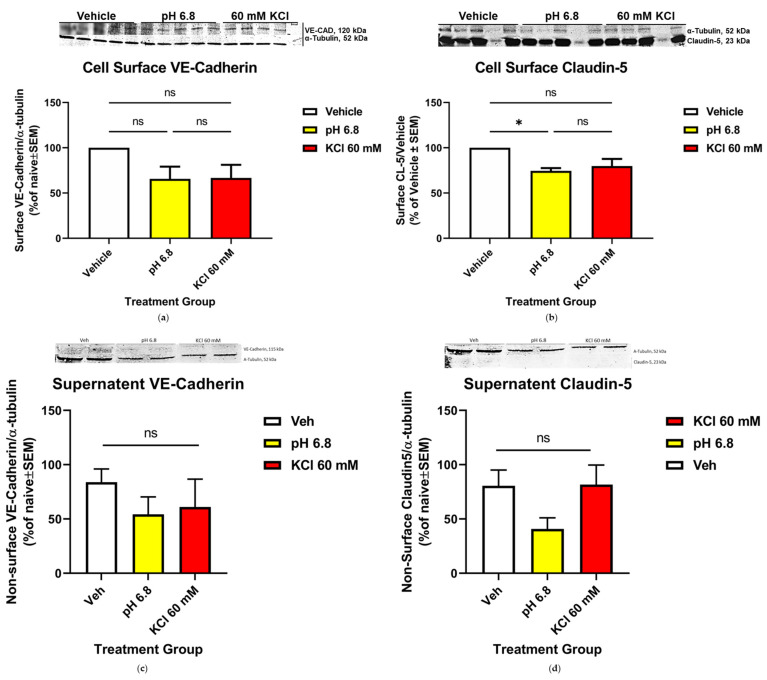
VE-CAD and CL-5 surface localization following KCl and acidified pH exposure in vitro. Biotynylation pulldown quantification plots of (**a**) VE-CAD and (**b**) CL-5 following 60 mM KCl and acidified pH insult. CL-5 detection was significantly increased following acidified pH exposure, while VE-CAD detection was not significantly affected, suggesting a direct pH-mediated mechanism on CL-5 localization, CL-5: Veh vs. 60 mM KCl: mean difference = 20.16, 95% CI (−0.8883, 41.21), *p* = 0.0587, n1 = 3, n2 = 3; Veh vs. pH = 6.8: mean difference = 25.49, 95% CI (4.444, 46.54), * *p* = 0.0231, n1 = 3, n2 = 3; analyzed by one-way ANOVA with Tukey’s multiple comparison test post hoc F(2,6) = 7.684, *p* = 0.0221. VE-CAD: Veh vs. 60 mM KCl: mean difference = 33.37, 95% CI (−16.46, 83.20), *p* = 0.1801, ns = no significance, n1 = 3, n2 = 3; Veh vs. pH = 6.8: mean difference = 34.35, 95% CI (−15.48, 84.17), *p* = 0.1668, n1 = 3, n2 = 3. Analyzed by one-way ANOVA with Tukey’s multiple comparison test post hoc F(2,6) = 2.900, *p* = 1315. No significant detection of VE-CAD (**c**) or CL-5 (**d**) was observed in supernatant collected during biotynylation (VE-CAD: Veh vs. 60 mM KCl: mean difference = 10.19, 95% CI (−153.7, 174.1), *p* = 0.9934, ns = no significance, n1 = 2, n2 = 2; Veh vs. pH = 6.8: mean difference = 25.14, 95% CI (−138.8, 189.1), *p* = 0.9191, ns = no significance. Analyzed by one-way ANOVA with Tukey’s multiple comparison test post hoc, F(3,4) = 0.1315, *p* = 0.9364; D, CL-5: Veh vs. 60 mM KCl: mean difference = 7.446, 95% CI (−227.8, 242.7), *p* = 0.9991, ns = no significance, n1 = 2, n2 = 2; Veh vs. pH = 6.8: mean difference = 65.03, 95% CI (−170.2, 300.2), *p* = >0.6957, ns = no significance. Analyzed by one-way ANOVA with Tukey’s multiple comparison test post hoc, F(3,4) = 1.366; *p* = 0.3732).

**Figure 4 pharmaceutics-14-01469-f004:**
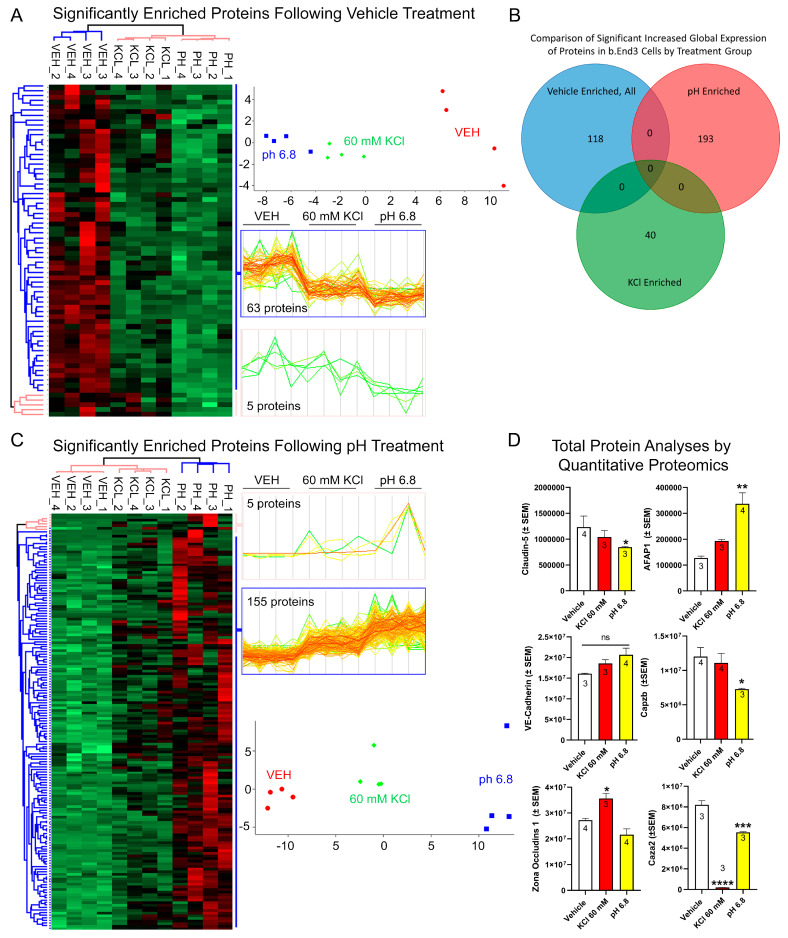
Global unlabeled quantitative proteomics analysis of b.End3 endothelial cells demonstrate convergent effect of independent mechanisms resulting from vehicle, 60 mM KCl, and media buffered to pH = 6.8 on expression of TJ and cytoskeletal proteins. (**A**) Unbiased hierarchal clustering and principal component analysis (PCA) of the 68 proteins enriched following vehicle treatment compared to 60 mM KCl and acidified pH obtained from 3-way ANOVA analysis (α = 0.05) of global proteome data illustrate consistency of protein expression between biological samples (*n* = 4 per treatment group) clustering accordingly into vehicle, 60 mM KCl, or pH = 6.8 buffered media treatment groups; heat map visualizes individual protein clustering. (**B**) Venn diagram comparing protein expression enrichment by treatment group, absence of proteins enriched by multiple treatment groups indicates unrelated mechanisms affecting protein expression for each treatment. (**C**) Unbiased hierarchal clustering and principal component analysis (PCA) of the 160 proteins enriched following acidified pH treatment compared to 60 mM KCl and vehicle obtained from 3-way ANOVA analysis (α = 0.05) of global proteome data illustrate consistency of protein expression between biological samples (*n* = 4 per treatment group) clustering accordingly into vehicle, 60 mM KCl, or pH = 6.8 buffered media treatment groups; heat map visualizes individual protein clustering. (**D**) Plots of TJ proteins CL-5, VE-CAD, ZO-1 and actin maintenance proteins AFAP1, Caza2, and Cap2b showing individual changes in total enrichment post KCl and acidified pH exposure, showing highly variable effects on total enrichment of structural proteins unique to each treatment and protein, indicative of a synergistic mechanism impairing barrier function summed from combined independent effects of each treatment (CL-5): vehicle vs. 60 mM KCl; mean difference = 192,265; 95% CI (−441,412, 825,941), *p* = 0.6393, *n* = 4/3: vehicle vs. pH = 6.8; mean difference = 387,221; 95% CI (−246,455, 1,020,898), *p* = 0.2286, *n* = 4/3; F(2,7) = 1.972, *p* = 0.3002: VE-CAD: vehicle vs. 60 mM KCl; mean difference = −2,479,175; 95% CI (−7,634,858, 2,676,509), *p* = 0.3617, *n* = 4/3: vehicle vs. pH = 6.8; mean difference = −4,557,060; 95% CI (−9,379,760, 265,640), *p* = 0.0946, *n* = 4/3; F(2,7) = 6.836, *p* = 0.0946: ZO-1: vehicle vs. 60 mM KCl; mean difference = −8,370,223; 95% CI (−15,234,252, −1,506,194), * *p* = 0.0207, ns = no significance; *n* = 4/3: vehicle vs. pH = 6.8; mean difference = −5,625,431; 95% CI (−729,425, 11,980,288), *p* = 0.0797, *n* = 4/3; F(2, 8) = 1.271, *p* = 0.0020: AFAP1: vehicle vs. 60 mM KCl; mean difference = −65,938; 95% CI (−192,646, 60,769) *p* = 0.3147, *n* = 4/3: vehicle vs. pH = 6.8; mean difference = −209,238; 95% CI (−327,762, −90714), ** *p* = 0.0034, *n* = 4/3; F(2, 7) = 58.55, *p* = 0.0047: Caza2: vehicle vs. 60 mM KCl; mean difference = 8,010,441; 95% CI (−6,881,290, 9,139,593) **** *p* = <0.0001, *n* = 3: vehicle vs. pH = 6.8; mean difference = 2,654,703; 95% CI (1,525,552, 3,783,855), *** *p* = 0.0006, *n* = 3; F(2, 7) = 194.7, *p* = <0.0001: Caz2b: vehicle vs. 60 mM KCl; mean difference = 911,433; 95% CI (−3,489,104, 5,311,970) *p* = 0.8099, *n* = 4/3: vehicle vs. pH = 6.8; mean difference = 4733908; 95% CI (−19,215, 9,487,031), *p* = 0.0508, *n* = 4/3; F(2,8) = 3.857, *p* = 0.0672. Analyzed by one-way ANOVA with Dunnett’s multiple comparison test post hoc.

**Figure 5 pharmaceutics-14-01469-f005:**
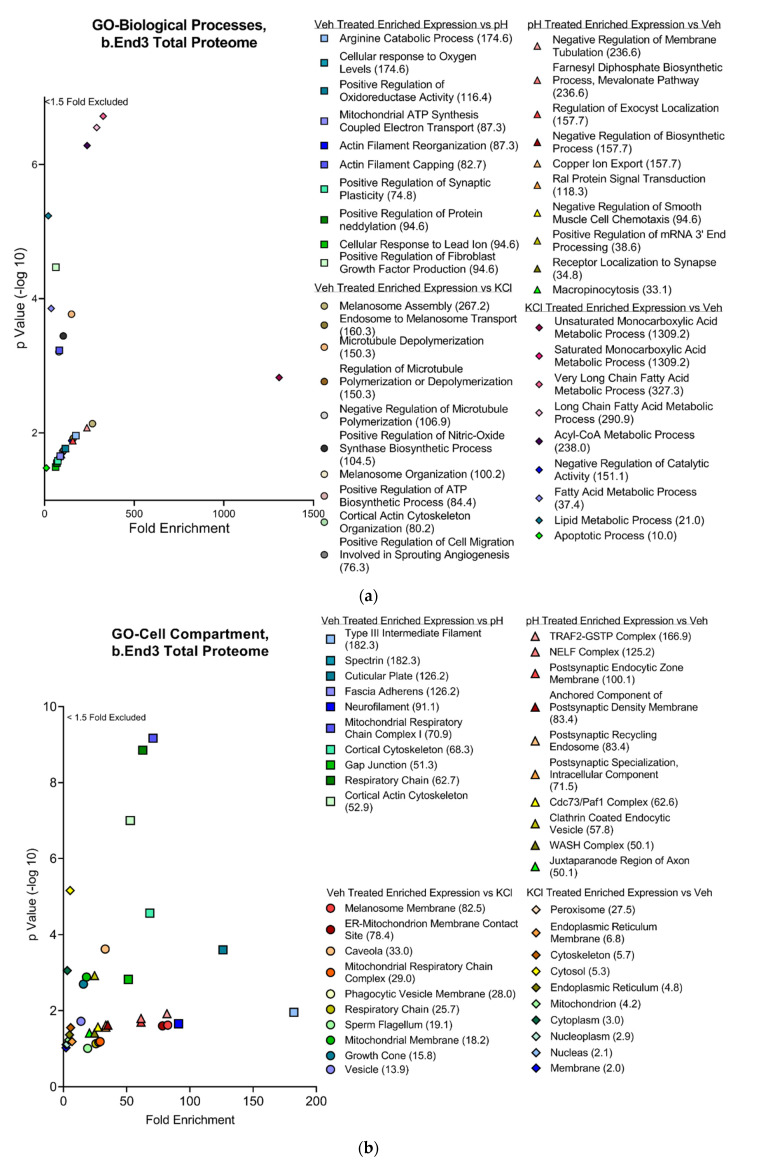
Global proteome significantly enriched proteins separated by treatment group and visualized with fold enrichment vs. −log10 transformed *p*-value scatterplots. Top ten cellular processes and pathways by total fold enrichment for the three GO databases (**a**) BP, (**b**) CC, (**c**) MF, and (**d**) KEGG pathway databases represented by individual plots demonstrate upstream convergence of changes in total protein enrichment of cell processes and pathways involved in maintenance of cell structural homeostasis. *n* = 4 per treatment group.

**Figure 6 pharmaceutics-14-01469-f006:**
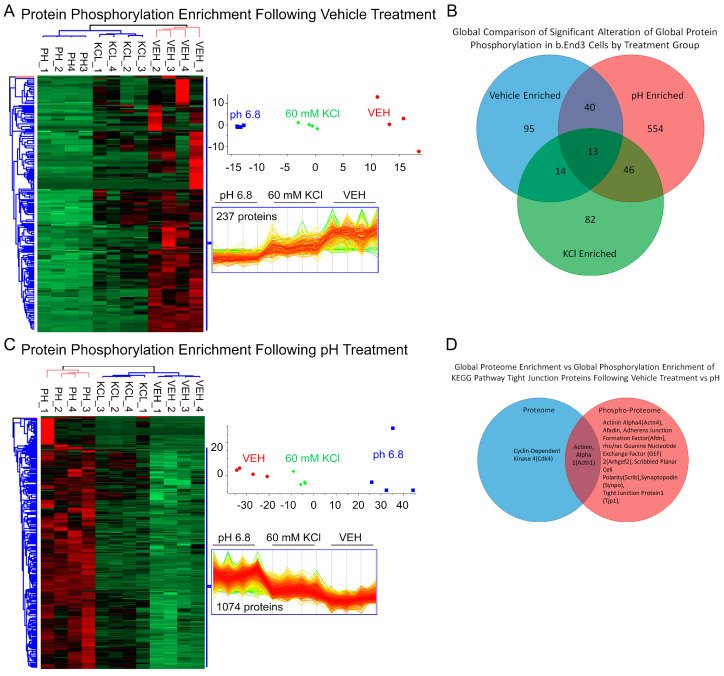
Global phospho-proteomic analysis of b.End3 endothelial cells following exposure to 60 mM KCl, acidified media buffered to pH = 6.8, and vehicle demonstrates variable phosphorylation enrichment of protein targets unique to each individual treatment, indicative of unique mechanisms of phosphorylation induction identifiable to each individual treatment. (**A**) Unbiased hierarchal clustering and principal component analysis of the 237 proteins experiencing phosphorylation enrichment following vehicle treatment when compared to KCl and acidified pH treatment, significance assessed by 3-way ANOVA (α = 0.05) analysis of global phospho-proteome data illustrating consistent phosphorylation enrichment between each biological sample (*n* = 4 per treatment group) clustering into vehicle, 60 mm KCl, and acidified media treatment groups. Heat map visualizes individual protein clustering. (**B**) Venn diagram comparing total protein phosphorylation enrichment by treatment group; enrichments can be largely identified with a single treatment, small degree of overlap due to large sample pool and ubiquitous nature of peptide phosphorylation. (**C**) Unbiased hierarchal clustering and principal component analysis of the 1074 proteins undergoing phosphorylation enrichment post exposure to acidified media compared to KCl and vehicle exposure; significance assessed by 3-way ANOVA (α = 0.05) analysis of global phospho-proteome data demonstrating consistent phosphorylation enrichment between biological samples (*n* = 4 per treatment group) with each protein clustering into vehicle, 60 mM KCl, and acidified media treatment groups. Heat map visualizes clustering of individual proteins. (**D**) Venn diagram comparing vehicle induced enrichment of total expression and phosphorylation enrichment of tight junction proteins identified in the KEGG pathway bioinformatic database, indicative of tandem alterations to PTM and overall expression of TJ proteins isolated to an individual treatment.

**Figure 7 pharmaceutics-14-01469-f007:**
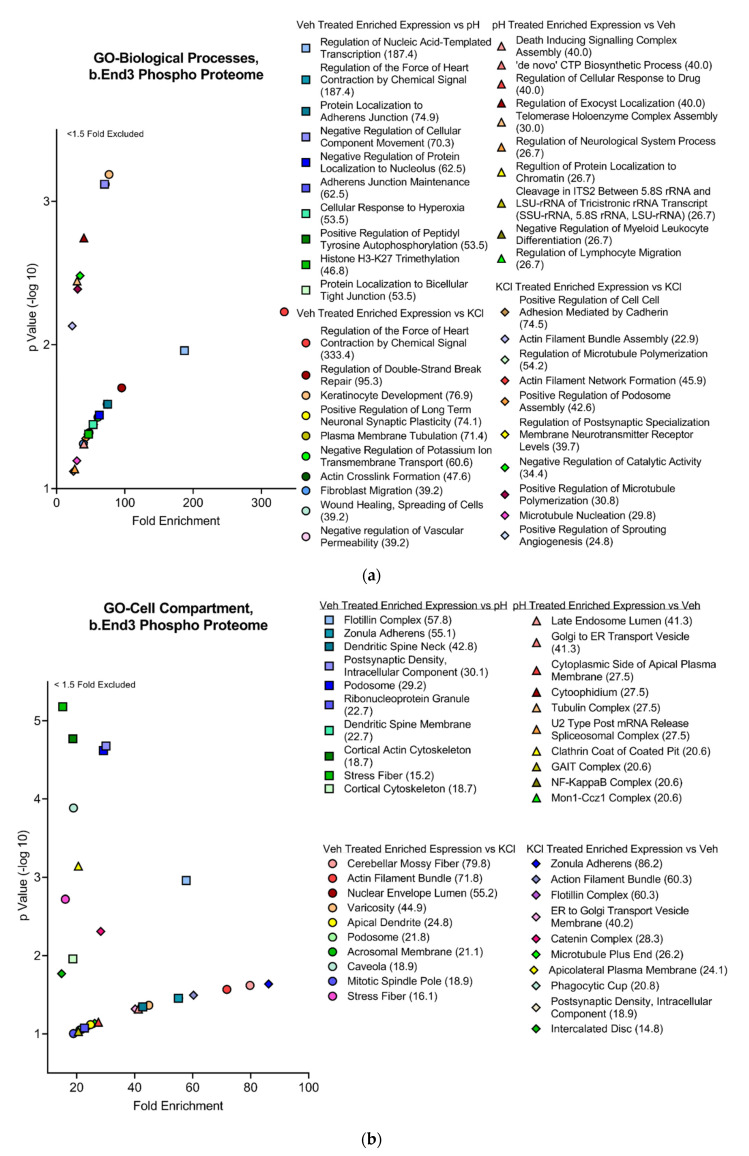
Global phosphorylation-enriched proteins separated by treatment group and represented by scatterplots comparing fold enrichment score (FE) against −Log10 transformed *p*-values. Top ten cellular processes and pathways assessed by total fold enrichment score queried through the three GO databases; (**a**) BP, (**b**) CC, (**c**) MF, and (**d**) KEGG pathway databases demonstrate highly varied upstream protein PTM modifications unique to each treatment group with convergence of effect observed in proteins associated with maintenance of structural integrity of the cell. *n* = 4 per treatment group.

**Figure 8 pharmaceutics-14-01469-f008:**
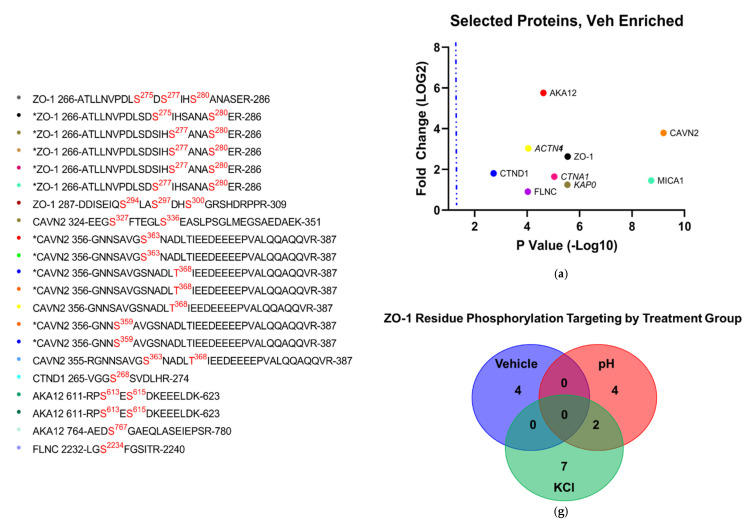
Scatterplots of significantly enriched phosphorylated proteins against physiologically associated proteins significantly enriched in phosphorylation following (**a**) vehicle, (**c**) acidified pH, and (**e**) 60 mM KCl insult. Individual fold change plots of phosphorylation-site enrichment, separated by (**b**) vehicle-enriched, (**d**) acidified pH-enriched, and (**f**) 60 mM KCl-enriched phosphorylation; individual dots on these plots correspond to listed verified detected PTM sites. When separated by treatment group, unique patterns of phosphorylation of ZO-1 can be identified by individual treatment, indicating PTM-regulated protein trafficking converges on this protein. Italicized proteins in scatterplots lacking verifiable phosphorylation sites were not included in individual fold change plots. Individual phosphorylated residues are colored red and superscript indicates position of phosphorylated amino acid; residue sequences marked with an asterisk indicate isobaric phosphorylation sites and were graphed as individual iterations of phosphopeptide enrichment. (**g**) Venn diagram illustrating overlap of phosphorylation adduct sites on ZO-1 residues by treatment group.

**Table 1 pharmaceutics-14-01469-t001:** Table of antibodies, vendor catalog number, dilution factor by application, and lot number.

Antibody	Vendor/Catalog Number	Application/Dilution	Lot Number
Claudin-5 Mouse mAb (4C3C2)	Invitrogen	WB: 1:500	WD
	35-2500	ICC: 1:50	327318
VE-Cadherin Rabbit pAb (CD144)	Invitrogen	WB: 1:500	UF
	36-1900	ICC: 1:200	287723
Zona Occludens-1 Mouse mAb (ZO1-1A12)	Invitrogen	ICC: 1:200	WG
	33-9100		329571
α-Tubulin (DM1A) Mouse mAb (DM1A)	Cell Signaling Technology	WB: 1:10,000	16
	3873		
Donkey anti-Rabbit IgG (H + L) Highly Cross-Adsorbed Secondary Antibody, Alexa Fluor^TM^ 488	Invitrogen A-21206	ICC: 1:10,000	2156521
Donkey anti-Mouse IgG (H + L) Highly Cross-Adsorbed Secondary Antibody, Alexa Fluor^TM^ 568	Invitrogen A-10037	ICC: 1:10,000	2156521
IRDye 800CW Donkey anti-Rabbit IgG Secondary Antibody	Li-Cor 926-32213	WB: 1:10,000	D10518-05
IRDye 680RD Donkey anti-Rabbit IgG Secondary Antibody	Li-Cor 926-68072	WB: 1:10,000	D10728-15
Alexa Fluor^TM^ 488 Phalloidin IR Dye 800CW Donkey anti-Rabbit IgG Secondary Antibody	Invitrogen A12379	WB: 1:10,000 ICC 1:40	D10518-052219253

## Data Availability

Gene ontology and Kyoto Encyclopedia of Genes and Genomes are publicly available through the DAVID bioinformatic website at https://david.ncifcrf.gov/. Data originally accessed 12 October 2021.

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
