# Peer review of "Extracellular Alterations in pH and K+ Modify the Murine Brain Endothelial Cell Total and Phospho-Proteome"

_pharmaceutics, 2022, doi:10.3390/pharmaceutics14071469_

Round 1

Reviewer 1 Report

This paper has a fatal flaw which makes it impossible for this reviewer to evaluate its scientific soundness. None of the western blot images follow the most basic guidelines for publication in scientific journals. Images are cropped and clearly a collage of different individual experiments. The quality of the tubulin bands (loading control for which a multitude of exceptionally good antibodies are available) are way below average. In order to be able to give a fair evaluation of this manuscript the quality of the western blot must be improved. 

Author Response

original blots have been uploaded 

Reviewer 2 Report

Cortical spreading depression/depolarization (CSD) is linked to many CNS diseases and the impaired BBB is one pathophysiological change. Substances released during CSD have vital roles in the progression of CSD and associated changes. Particularly, extracellular elevated K+, low pH, glutamate, and ATP are common detected substances from CSD. In this study, K+ ion and low pH had been identified active in altering endothelial paracellular communications in bEnd.3 cell model. Claudin 5 was demonstrated to be target by the treatments of K+ and low pH through membrane relocalization. Additionally, using proteomic analysis, post translational modification and ZO-1 were revealed to be involved.

  1. Sadeghian et al. (Ann Neurol 2018;84:409-423) investigated the effects of CSD on BBB integrity in mice model. Endothelial transcytosis, instead of changes in cell types and TJ proteins, through caveolin-1 dependent mechanism was reported attributable to CSD-induced BBB permeability change. Current in vitro cell model findings differed with data of in vivo. The citation of this reference and comparison are essential to the improved manuscript.
  2. In vivo, endothelial cells were closely contacted and interacted with cell types and base membrane. The effects and roles of factors other than endothelial cells should be taken into consideration.
  3. The establishment of endothelial monolayer in bEnd.3 cells was supplemented with ACM. Why? The preparation and rationale should be described.
  4. The formation and integrity of endothelia cell monolayer are regulated by multiple factors, including matrix, cell-cell contact, divalent ions, and pH. This experimental condition used in this study varied with in vivo.
  5. A conclusive finding regarding proteomic study and implication should be clearly made.

Author Response

Reviewer 2

Cortical spreading depression/depolarization (CSD) is linked to many CNS diseases and the impaired BBB is one pathophysiological change. Substances released during CSD have vital roles in the progression of CSD and associated changes. Particularly, extracellular elevated K+, low pH, glutamate, and ATP are common detected substances from CSD. In this study, K+ ion and low pH had been identified active in altering endothelial paracellular communications in bEnd.3 cell model. Claudin 5 was demonstrated to be target by the treatments of K+ and low pH through membrane relocalization. Additionally, using proteomic analysis, post translational modification and ZO-1 were revealed to be involved.

  1. Sadeghian et al. (Ann Neurol 2018;84:409-423) investigated the effects of CSD on BBB integrity in mice model. Endothelial transcytosis, instead of changes in cell types and TJ proteins, through caveolin-1 dependent mechanism was reported attributable to CSD-induced BBB permeability change. Current in vitro cell model findings differed with data of in vivo. The citation of this reference and comparison are essential to the improved manuscript.

Citation of Sadeghian et al has been added to the Discussion section, in addition to a comparison with both the in vitro studies undertaken in this manuscript, as well as to approaches undertaken in previous studies of the BBB and CSD in in vivo models of migraine by previous members of our lab group (Cottier et al., 2018).

  1. In vivo, endothelial cells were closely contacted and interacted with cell types and base membrane. The effects and roles of factors other than endothelial cells should be taken into consideration.

Utilization of co-culture in transwell system of endothelial cells with astrocyte conditioned media was used to model the dynamic interactions and essential roles of the various cell types found within the neurovascular unit, specifically pertaining to maintenance of proper barrier properties.  At the outset of these studies media conditioned in both neurons and astrocytes was attempted but proved to be unworkable with the resources available to us at the time.  Pericytes were also considered, however attempts to culture them proved difficult under our conditions.  Future investigations utilizing this model will include usage of human cell lines and potential use of triple contact culture for better in vitro modelling of the BBB.  Further in vivo studies in our lab seek to assay this issue with in vitro modelling of the BBB.

  1. The establishment of endothelial monolayer in bEnd.3 cells was supplemented with ACM. Why? The preparation and rationale should be described.

Astrocyte conditioned media (ACM) was utilized in this study to better create an in vitro model of the BBB by facilitating formation of a functional endothelial monolayer in our culture system.  Preparation of ACM was explained in cell treatment section of Materials and Methods for clarity. 

  1. The formation and integrity of endothelia cell monolayer are regulated by multiple factors, including matrix, cell-cell contact, divalent ions, and pH. This experimental condition used in this study varied with in vivo.

Preliminary in vitro studies undertaken in our lab had elucidated the divergent findings between the in vivo and in vitro models of the BBB response to CSD insult.  These divergent findings were discussed in response to comment one in the discussion section (please see response to comment 1 above.)  As previously stated, in vivo findings conducted in tandem with these studies have been undertaken in our lab (please see Cottier et al and Palomino et al) and reporting of these data were relegated to this separate investigation.  As per comment one, further elaboration on these experimental conditions have been added to the discussion section.

  1. A conclusive finding regarding proteomic study and implication should be clearly made.

Findings and conclusions drawn from total proteome arm of this study, and implications to the total investigation has been further expounded upon in discussion section of the manuscript.

Reviewer 3 Report

The manuscript entitled ‘Extracellular alterations in pH and K+ modify the murine brain endothelial cell’ by Wahl J. et al represents a very interesting manuscript where the authors investigated the alterations on the BEB in vitro induced by substances released during CSD which were assayed by TEER. They showed that relocalization and functional alteration to proteins associated with the cytoskeleton and endothelial tight junctions. Interestingly, the authors found to have unique phosphorylation signatures in phospho-proteome analysis, identifying Zona Occludens 1 as a possible pathologic “checkpoint” of the BBB.

In overall, I consider that the premise of this study is very interesting and important for the field, and I will perform some comments and suggestions.

Major and Minor concerns:

  1. In the following statements the authors are referring to actin cytoskeleton? They should specify actin cytoskeleton ‘Findings demonstrated relocalization and functional alteration to proteins associated with the cytoskeleton and endothelial tight junctions’ (abstract)
  2. A brief explanation of TEER is missing in my opinion.
  3. In the figure 1B, the statistics of disodium-ATP (100 uM) condition is missing.
  4. In the Figure 2, separate images of both channels detecting both claudin5 and VE cadherin should be presented.
  5. The following statement is not correct, and should be rephrased ‘Changes in total protein expression were further assessed with quantitative proteomics’. The authors only evaluated the proteome and the phosphoproteome and not expression levels.
  6. With the images presented in Figure 2, is very difficult observe an increase sin tress fibers. The authors should quantify the presence of stress fibers in each condition and rephrase the following statement. ‘When compared to vehicle, an apparent increase in stress fibers can be observed in both KCl and acidified pH treatments, indicating insult to normal structural homeostasis of the cell´.
  7. Table 1 should be included in the manuscript.
  8. The immunoblots presented in the figure 3 are of poor quality and unacceptable. The loading control a-tubulin present higher molecular weight in 2 condition tested when compared with control (vehicle) (figure 3A and B). Why? The molecular weights should be included in the immunoblots.
  9. Why a-tubulin is changing with your treatments (Figure 3C and D)?
  10. The figure 5 and Figure 7 are difficult to read.
  11. Figure 8B, D, F are impossible to analyze.
  12. Why two legends are presented in each figure? (figure 1, Figure 2, Figure 3, Figure 4,…all of them).
  13. Please avoid the abbreviations and when needed please, write them in full before abbreviation (example TEER screening (abstract), TBI, MS (introduction).

Author Response

Reviewer 3

The manuscript entitled ‘Extracellular alterations in pH and K+ modify the murine brain endothelial cell’ by Wahl J. et al represents a very interesting manuscript where the authors investigated the alterations on the BEB in vitro induced by substances released during CSD which were assayed by TEER. They showed that relocalization and functional alteration to proteins associated with the cytoskeleton and endothelial tight junctions. Interestingly, the authors found to have unique phosphorylation signatures in phospho-proteome analysis, identifying Zona Occludens 1 as a possible pathologic “checkpoint” of the BBB.

In overall, I consider that the premise of this study is very interesting and important for the field, and I will perform some comments and suggestions.

Major and Minor concerns:

  1. In the following statements the authors are referring to actin cytoskeleton? They should specify actin cytoskeleton ‘Findings demonstrated relocalization and functional alteration to proteins associated with the cytoskeleton and endothelial tight junctions’ (abstract)

Cytoskeleton reference in abstract corrected to “actin cytoskeleton”

  1. A brief explanation of TEER is missing in my opinion.

Brief explanation of principle and application of TEER added to materials and methods section

  1. In the figure 1B, the statistics of disodium-ATP (100 uM) condition is missing.

Statistics for 100 μM disodium-ATP added to Fig 1B figure legend

  1. In the Figure 2, separate images of both channels detecting both claudin5 and VE cadherin should be presented.

Separate channels for CL-5 and VE-CAD immunofluorescence added to panel of microscopy images in figure 2A

  1. The following statement is not correct and should be rephrased ‘Changes in total protein expression were further assessed with quantitative proteomics’. The authors only evaluated the proteome and the phosphoproteome and not expression levels.

Statement corrected to “Alterations to the global b.End3 proteome following KCl or acidic pH insult were then assessed with quantitative proteomics” in section 3.2

  1. With the images presented in Figure 2, is very difficult observe an increase sin tress fibers. The authors should quantify the presence of stress fibers in each condition and rephrase the following statement. ‘When compared to vehicle, an apparent increase in stress fibers can be observed in both KCl and acidified pH treatments, indicating insult to normal structural homeostasis of the cell´.

Corrected Total Cell Fluorescence (CTCF) quantification performed and added to figure 2, phrasing corrected to “Following KCl insult, a significant increase in F-actin stress fibers were observed when compared to vehicle (Fig 2A-B), indicative of potential homeostatic stress response following KCl insult.” In section 3.2.2

  1. Table 1 should be included in the manuscript.

Table 1 has been added to materials and methods section, page 5

Table one has been added to page

  1. The immunoblots presented in the figure 3 are of poor quality and unacceptable. The loading control a-tubulin present higher molecular weight in 2 condition tested when compared with control (vehicle) (figure 3A and B). Why? The molecular weights should be included in the immunoblots.

Immunoblots in Fig 3 A and B were performed again, vehicle was used as a control for quantification of pH and KCl treatment groups for cell surface Claudin-5 detection, while VE-CAD was quantified with tubulin.  High detection levels of surface proteins vs intracellular (tubulin) or adherens junction proteins (VE-CAD) expected for this type of assay.  Figures 3 C-D were removed due to poor quality of immunoblots, with CL-5 not showing any qualitative detection.

  1. Why a-tubulin is changing with your treatments (Figure 3C and D)?

Tubluin detection in these figures was not changing after followup up experiments to address reviewer 1.  Figure 3C and D were removed for this resubmission.

  1. The figure 5 and Figure 7 are difficult to read.

All figures in manuscript converted to high resolution TIFF files and only one is present per page enabling ease of reading

  1. Figure 8B, D, F are impossible to analyze.

See reply to comment 10

  1. Why two legends are presented in each figure? (figure 1, Figure 2, Figure 3, Figure 4,…all of them).

Duplicate figure legend titles removed for all figures in manuscript.

  1. Please avoid the abbreviations and when needed please, write them in full before abbreviation (example TEER screening (abstract), TBI, MS (introduction).

All abbreviations fully written and defined before and use in the body of the manuscript

Reviewer 4 Report

Authors used a murine model of blood endothelial barrier by using of bEnd.3 murine immortalized endothelial cells grown in astrocyte conditioned media to investigate the effects of elevation of potassium levels (60 mM) and acidification of pH (6.8), processes that occur in neurological disorders as cortical spreading disease, on the paracellular integrity, expression, and localization of tight junction proteins. Additionally, they analyze the changes in protein phosphorylation with the purpose of found and characterize specific molecular changes that can be utilized as a diagnostic fingerprint to identify determined pathology and thereby tentatively apply a certain treatment.

To achieve these goals they used diverse techniques and maneuvers, properly chosen, to analyze the transport process evaluated by trans-endothelial electrical resistance, and 14C-sucrose, and 4- and 70 KDa fluorescein isothiocyanate-dextran uptake assays.

Western blot, biotinylation assays, confocal microscopy and immunofluorescent techniques were also used. Proteomics and phospho-proteomics analyses were made also in this study to analyze relocalization of the tight juntion proteins.

They conclude that the aforementioned alterations in potassium levels and pH values induced alterations in tight junctions as well as actin cytoskeletal reorganization that affect the functions of blood endothelial barrier and blood brain barrier.

This is a very well done job that involved a lot of work. The methods used are described in detail and the results obtained are clearly showed.

Minor observations.

Please:

Provide the meaning for all abbreviations used in the text.

Write properly the concentration units (for L-glutamine and for other substances) as well as “SDS-Page” abbreviation and “phosho-proteome analysys” phrase.

Observe that there are words inappropriately spelled with the first letter capitalized as well as separated by a space (examples: “mode ling”, “Pris m”).

Separate some numbers and concentration units (example: 100µM).

Correct the full stop for references of Andrew, Hsieh, and Brisson 2017 (page 3) as well as Kruse et al. 2017 (page 5).

A parenthesis is missing for Alexa Fluor secondary antibodies data (brand and catalogue number).

Correct the brand name “PerkinsElmer” and the volume unit for microfuge tube used for protein analysis.

Clarify the time for washes at 4oC of neutravidin beads in biotinylation assays.

Choose how write the cells used: “bEnd.3 cells” or “bEND.3 cells”.

Clarify if FITC is the abbreviation for fluorescence isothiocyanate-dextran or for fluorescein isothiocyanate-dextran.

Provide the catalogue number and brand for 70 KDa-FITC.

Include the meaning for CTCF in legend of figure 2.

Author Response

Reviewer 4

Authors used a murine model of blood endothelial barrier by using of bEnd.3 murine immortalized endothelial cells grown in astrocyte conditioned media to investigate the effects of elevation of potassium levels (60 mM) and acidification of pH (6.8), processes that occur in neurological disorders as cortical spreading disease, on the paracellular integrity, expression, and localization of tight junction proteins. Additionally, they analyze the changes in protein phosphorylation with the purpose of found and characterize specific molecular changes that can be utilized as a diagnostic fingerprint to identify determined pathology and thereby tentatively apply a certain treatment.

To achieve these goals they used diverse techniques and maneuvers, properly chosen, to analyze the transport process evaluated by trans-endothelial electrical resistance, and 14C-sucrose, and 4- and 70 KDa fluorescein isothiocyanate-dextran uptake assays.

Western blot, biotinylation assays, confocal microscopy and immunofluorescent techniques were also used. Proteomics and phospho-proteomics analyses were made also in this study to analyze relocalization of the tight juntion proteins.

They conclude that the aforementioned alterations in potassium levels and pH values induced alterations in tight junctions as well as actin cytoskeletal reorganization that affect the functions of blood endothelial barrier and blood brain barrier.

This is a very well done job that involved a lot of work. The methods used are described in detail and the results obtained are clearly showed.

Minor observations.

Please:

Provide the meaning for all abbreviations used in the text.

All abbreviations fully written and defined before and use use in the body of the manuscript (see response to comment 13 from reviewer 3)

Write properly the concentration units (for L-glutamine and for other substances) as well as “SDS-Page” abbreviation and “phosho-proteome analysis” phrase.

Misspellings of abbreviations and concentration units have been corrected in the body of the manuscript.

Observe that there are words inappropriately spelled with the first letter capitalized as well as separated by a space (examples: “mode ling”, “Pris m”).

Manuscript edited and spelling/grammar errors have been corrected.

Separate some numbers and concentration units (example: 100µM).

Concentration units and numbers corrected to standardized forms

Correct the full stop for references of Andrew, Hsieh, and Brisson 2017 (page 3) as well as Kruse et al. 2017 (page 5).

All references used in the manuscript have been corrected and reformatted according to the MDPI manuscript formatting guide

A parenthesis is missing for Alexa Fluor secondary antibodies data (brand and catalogue number).

Parenthesis added to Alexa Flour secondary antibody data in the manuscript

Correct the brand name “PerkinsElmer” and the volume unit for microfuge tube used for protein analysis.

Brand name corrected to PerkinElmer, volume unit corrected to 1.7 ml microfucge tube

Clarify the time for washes at 4oC of neutravidin beads in biotinylation assays.

Neutravidin bead wash procedure rewritten for clarification in materials and methods section under “biotynylation” heading

Choose how write the cells used: “bEnd.3 cells” or “bEND.3 cells”.

All usage of the b.End.3 abbreviation in the manuscript standardized to “b.End.3”

Clarify if FITC is the abbreviation for fluorescence isothiocyanate-dextran or for fluorescein isothiocyanate-dextran.

All references to FITC in the manuscript has been standardized to “fluorescein isothiocyanate-dextran.

Provide the catalogue number and brand for 70 KDa-FITC.

Catalogue number and brand of 70 kDa FITC added to FITC transport assay section of materials and methods.

Include the meaning for CTCF in legend of figure 2.

Corrected Total Cell Fluorescence (CTCF) added to figure 2 legend to clarify use of abbreviation

  1. Graphical Abstract was not provided. Kindly provide one in the
    correct format (PNG or JPEG, height x width 1000 x 600)

Graphical abstract has been added to the discussion portion of the manuscript
2.      Figures provided were of too low resolution (ie Fig 7. Words too small to be seen). Kindly revise.

All figures in the entire manuscript have been replaced with high resolution TIFF images (Please see response to comment 10 by reviewer 3 above)

  1.     Manuscript was not formatted according to MDPI template

Entirety of manuscript has been reformatted according to MDPI formatting guidelines

  1.     References are not formatted properly.

References used within this manuscript have been reformatted according to MDPI formatting guide
5.      Kindly provide originals for western blots.

All original western blots utilized within this manuscript have been provided as a supplementary figure (Fig S1) at the end of the manuscript.

Round 2

Reviewer 2 Report

None.

Reviewer 3 Report

Please find my comments about the revision of the manuscript entitled Extracellular alterations in pH and K+ modify the murine brain endothelial cell’ by Wahl J. et al. The authors respond to all issues raised in my revision and performed the adequate alterations of the manuscript and the latter was improved.

Overall, I believe that the manuscript is ready for publication.